

# River effects on sea-level rise in the Río de la Plata during the past century

Christopher G. Piecuch[1]

[1]Woods Hole Oceanographic Institution, Woods Hole, Massachusetts, USA

**Correspondence:** Christopher G. Piecuch (cpiecuch@whoi.edu)

**Abstract.** Identifying the causes for historical sea-level changes in coastal tide-gauge records is important for constraining oceanographic, geologic, and climatic processes. The Río de la Plata estuary in South America features the longest tide-gauge records in the South Atlantic. Despite the relevance of these data for large-scale circulation and climate studies, the mechanisms underlying relative sea-level changes in this region during the past century have not been firmly established. I study annual data from tide gauges in the Río de la Plata and stream gauges along the Río Paraná and Río Uruguay to establish relationships between river streamflow and sea level over 1931–2014. Regression analysis suggests that streamflow explains $59\% \pm 17\%$ of the total sea-level variance at Buenos Aires, Argentina, and $28\% \pm 21\%$ at Montevideo, Uruguay (95% confidence intervals). A longterm streamflow increase effected sea-level trends of $0.71 \pm 0.35$ mm yr$^{-1}$ at Buenos Aires and $0.48 \pm 0.38$ mm yr$^{-1}$ at Montevideo. More generally, sea level at Buenos Aires and Montevideo respectively rises by $(7.3 \pm 1.8) \times 10^{-6}$ m and $(4.7 \pm 2.6) \times 10^{-6}$ m per 1 m$^3$ s$^{-1}$ streamflow increase. These observational results are consistent with simple theories for the coastal sea-level response to streamflow forcing, suggesting a causal relationship between streamflow and sea level mediated by ocean dynamics. Findings advance understanding of local, regional, and global sea-level changes, clarify sea-level physics, inform future projections of coastal sea level and the interpretation of satellite data and proxy reconstructions, and highlight future research directions.

## 1 Introduction

Tide-gauge records of relative sea level go back more than a century in some places, representing some of the longest instrumental time series of the Earth system (Hogarth, 2014; Talke et al., 2018; Woodworth et al., 2010). On long climate time scales, changes in global-mean sea level are informative of global ocean warming, land ice wastage, and terrestrial water storage, whereas local and regional deviations from the global average shed light on processes including ocean dynamics and gravitation, rotation, and solid-Earth deformation (Gregory et al., 2019; Horton et al., 2018; Kopp et al., 2015). Identifying the mechanisms responsible for sea-level changes observed in tide-gauge records is therefore a major goal in geophysics, oceanography, and climate science (Douglas et al., 2001; Emery and Aubrey, 1991; Lisitzin, 1974).

The nature and causes of twentieth-century sea-level changes in the South Atlantic Ocean are poorly understood compared to behavior in other ocean basins during the same time period (Dangendorf et al., 2017; Frederikse et al., 2018). This knowledge gap reflects a lack of data—the basin has few long tide-gauge records (Hamlington and Thompson, 2015; Natarov et al., 2017).



Given the basin's large area (Thompson and Merrifield, 2014), the absence of long data records in the South Atlantic Ocean poses a particular challenge to estimates of global-mean sea-level rise (Church and White, 2011; Dangendorf et al., 2017; Frederikse et al., 2020; Hay et al., 2015; Jevrejeva et al., 2014; Ray and Douglas, 2011), but also to our understanding of circulation and climate during the past century more generally.

A recent study brings together available tide-gauge records along with other data, proxies, and models to quantify rates and mechanisms of twentieth-century South-Atlantic sea-level change (Frederikse et al., 2021). Those authors determine that sea level in the South Atlantic rose about 0.3 mm yr$^{-1}$ faster than the rate of global-mean sea-level rise, owing to a combination of ocean dynamics and gravitational, rotational, and deformational effects from contemporary mass redistribution. Importantly, their estimate of twentieth-century sea-level rise over the South Atlantic rests heavily on a handful of long tide-gauge records

in and around the Río de la Plata, which feature large sea-level trends that have been reported on previously (Aubrey et al., 1988; Brandani et al., 1985; D'Onofrio et al., 2008; Dennis et al., 1995; Douglas, 1997, 2001, 2008; Emery and Aubrey, 1991; Fiore et al., 2009; Isla, 2008; Lanfredi et al., 1988, 1998; Melini et al., 2004; Pousa et al., 2007; Verocai et al., 2016).

The Río de la Plata is a long, broad, shallow salt-wedge estuary that widens from $\sim 50$ km to $\sim 250$ km and deepens from $\sim 5$ m to $\sim 20$ m between Buenos Aires, Argentina and Punta del Este, Uruguay, before emptying out onto the shelf (Guerrero

et al., 1997; Verocai et al., 2016; Figures 1, 2). The estuary is typified by a strong salinity and turbidity front at Barra del Indio Shoal between Punta Piedras, Argentina and Montevideo, Uruguay, with fresher, more turbid waters upstream to the northwest, and saltier, less turbid waters downstream to the southeast (Acha et al., 2018; Guerrero et al., 1997; Moreira and Simionato, 2019). These features, and the region's hydrography and ecology generally, are strongly shaped by the situation of the estuary at the confluence of the Río Paraná and Río Uruguay, which are two of the world's largest rivers by streamflow and drainage.

Streamflow into the Río de la Plata is known to have increased in the past century (Dai, 2016; Dai and Trenberth, 2002; Dai et al., 2009; cf. Figure 3). However, the possible influence of the increased streamflow on multidecadal and centennial sea-level trends remains largely unexplored. Discussions of the connection between streamflow and regional sea level are mostly qualitative, and center on interannual variability at Buenos Aires in relation to El Niño; for example, precipitation over the Plata Basin, streamflow of the Río Paraná and Río Uruguay, and sea level at Buenos Aires tend to increase in succession during

El Niño events (Douglas, 2001; Frederikse et al., 2021; Isla, 2008; Meccia et al., 2009; Papadopoulous and Tsimplis, 2006; Raicich, 2008; Santamaria-Aguilar et al., 2017; Thompson et al., 2016; Verocai et al., 2016). Douglas (2001) and Thompson et al. (2016) argue that sea-level trends calculated from the Buenos-Aires tide gauge are effected by sea-level variability during the 1982–1983 El Niño. While both studies relate this variability to river effects, Douglas (2001) favors an interpretation in terms of ocean dynamics, whereas Thompson et al. (2016) appeal to gravitational, rotational, and deformational effects. Alternative

interpretations of regional tide-gauge trends are given by Aubrey et al. (1988) and Melini et al. (2004) generally in terms of continental crustal rifting and subsidence, and the sea-level response to the 1960 Valdivia earthquake, respectively. Therefore, it remains unclear what processes mediate the relationship between streamflow and sea level, how these two variables are related more broadly as a function of time, and whether such considerations are relevant for interpreting longterm sea-level trends. Needed is a dedicated comparison of long stream- and tide-gauge records that provides a physical interpretation and

establishes causality.





Did streamflow effect longterm sea-level trends at tide gauges in the Río de la Plata? If so, what processes were involved? To answer these questions, I apply statistical analyses to annual data from stream gauges and tide gauges over the past century, and I formulate simple theories based on ocean dynamics to interpret the results. I conclude that local estuarine and coastal ocean dynamics forced by changes in streamflow had an important impact on twentieth-century sea-level rise in the Río de la Plata. Once adjusted for these effects and background late-Holocene rates, both of which contribute negligibly to changes in global-ocean water volume, the tide gauges show trends more in line with contemporary estimates of twentieth-century global-mean sea-level rise (Dangendorf et al., 2017; Frederikse et al., 2020; Hay et al., 2015). The remainder of this paper is structured as follows: in section 2, I describe the datasets; I report on results of the observational analysis, which involves correlation and regression methods applied to the data, in section 3; in section 4, I develop simple analytical models of the sea-level response to streamflow forcing to interpret observational results from section 3; finally, I conclude with a summary and discussion in section 5.

## 2 Data

### 2.1 Streamflow

I use yearly streamflow records from the Global Streamflow Indices and Metadata Archive (GSIM; Do et al., 2018; Gudmundsson et al., 2018). The GSIM database gives data from 3 stream gauges along the Río Paraná and 12 from the Río Uruguay (Table 1; Figure 3). To estimate Río de la Plata streamflow, I combine data from the two rivers. Records from the Río Paraná are long and complete. Therefore, I use the time series from Timbúes, which spans 1905–2014 and has the largest gauged area. Data from the Río Uruguay are shorter and more gappy; for example, the station with the largest drainage, Aporte Salto Grande, only gives data for 2012–2016. Since drainage area and mean streamflow are strongly correlated across stream gauges along this river (Pearson correlation coefficient $> 0.99$; Table 1), I create a composite streamflow time series for the Río Uruguay covering 1931–2016 by averaging the available records after scaling each station's time series by the ratio of the total drainage area to the drainage monitored by that particular gauge. Summing the Río Paraná data at Timbúes and the composite Río Uruguay record gives a complete time series of Río de la Plata streamflow for 1931–2014 (Figure 3).

### 2.2 Relative sea level

I use annual relative sea level records from the Permanent Service for Mean Sea Level (PSMSL; Holgate et al., 2013; PSMSL, 2022). The PSMSL database extracted on 21 March 2022 provides long ($> 50$-year) time series reduced to a common datum for 6 tide gauges from three regions in and around the Río de la Plata: Buenos Aires and Palermo towards the head of the estuary in Argentina; Montevideo and La Paloma near the mouth of the estuary along the coast of Uruguay to the north; and Mar del Plata and Quequén outside of the estuary along coastal Argentina to the south (Table 2; Figures 1, 4). To extend record length and reduce dimensionality, I average adjacent pairs of tide-gauge records relative to their common period, creating longer virtual-station records (Dangendorf et al., 2017; Frederikse et al., 2021; Jevrejeva et al., 2014) at Buenos Aires (1905–



2019), Montevideo (1938–2018), and Mar del Plata (1918–2019). For each station, I interrogate the period of overlap between virtual-station and stream-gauge data.

## 2.3 Late-Holocene trends

To distinguish late-Holocene trends related to background geological processes from modern rates of change due to ocean circulation and climate in the tide-gauge records, I use proxy reconstructions of relative sea level from Santa Catarina, Brazil compiled by Milne et al. (2005) and originally reported by Angulo et al. (1999) based on Vermetid snails (Table 3). These mollusks are sea-level indicators because they grow formations between the infra- and midlittoral zones, so formations fossilized in growth position are informative of low water (Laborel, 1986). Applying Bayesian linear regression to the data, and accounting

for the relative sea level and age errors, I determine a relative sea-level trend during the past 2,000 years of $-0.54 \pm 0.32$ mm yr$^{-1}$ (95% posterior credible interval); the Bayesian model is detailed in the Appendix. This negative rate of change arises from ocean siphoning and continental levering (Mitrovica and Milne, 2002), and past modeling studies of the glacial isostatic adjustment process report similar rates over the past few millennia (Caron et al., 2018; Peltier, 2004).

## 3 Results

Mean Río de la Plata streamflow is $(2.2 \pm 0.1) \times 10^4$ m$^3$ s$^{-1}$ (Figure 3), which is one of the largest river flows in the world, and consistent with values in past studies (Guerrero et al., 1997). Unless otherwise indicated, $\pm$ values identify 95% bootstrap confidence intervals. The record standard deviation of $(4.8 \pm 1.0) \times 10^3$ m$^3$ s$^{-1}$ quantifies variability across interannual to multidecadal time scales, including a longterm trend of $96 \pm 37$ m$^3$ s$^{-1}$ yr$^{-1}$, which has been reported on previously (Dai, 2016; Dai et al., 2009). Interannual variations in streamflow partly correspond to El Niño Southern Oscillation (ENSO); the

correlation coefficient between streamflow and the Niño 3.4 Index (Rayner et al., 2003) is $0.33 \pm 0.16$, and peak streamflow occurred during the 1982–1983 and 1997–1998 El Niños. Such relationships between streamflow and ENSO have been extensively documented (Berri et al., 2002; Cardoso and Silva Dias, 2006; Depetris et al., 1996; Grimm et al., 1998; Robertson and Mechoso, 1998; Ropelewski and Halpert, 1987). Also apparent is a regime shift from the late 1960s to early 1980s when streamflow increased substantially. This transition has been ascribed to increased precipitation and decreased evaporation over

the drainage basin due to changes in land use, deforestation, and large-scale climate modes (Lawrence and Vandecar, 2015; Medvigy et al., 2011).

The virtual-station data similarly show that relative sea level varies over all periods (Figure 4). These records also exhibit spatial structure. Detrended series at Buenos Aires and Montevideo are significantly correlated with one another (correlation coefficient $0.44 \pm 0.20$), but neither is correlated with the detrended record at Mar del Plata (coefficients $-0.01 \pm 0.20$ and

120 $0.18 \pm 0.28$, respectively). While the time series at Mar del Plata is uncorrelated with ENSO (correlation coefficient $0.12 \pm 0.17$ with Niño 3.4), the records from Buenos Aires and Montevideo both show correlation with ENSO (coefficients $0.26 \pm 0.19$ and $0.25 \pm 0.20$ with Niño 3.4, respectively). These results are consistent with past studies (Douglas, 2001; Papadopoulous and Tsimplis, 2006; Raicich, 2008; Verocai et al., 2016), and suggest that there exist processes that drive common sea-level





changes at Buenos Aires and Montevideo, but which do not effect sea level along Mar del Plata. Considering the longest time
scales, I compute a longterm rate of change at Buenos Aires of $1.46 \pm 0.36$ mm yr$^{-1}$ based on ordinary least squares linear
regression, which is larger than the trends of $1.03 \pm 0.53$ and $1.00 \pm 0.35$ mm yr$^{-1}$ obtained for Montevideo and Mar del Plata,
respectively (Figure 7). These values agree with previous studies of regional sea-level rise, cited in the introduction. After
adjusting for a late-Holocene rate (section 2.c; Figure 5), I find an average sea-level trend across virtual stations of $1.70 \pm 0.40$
mm yr$^{-1}$, which is faster than modern estimates of twentieth-century global-mean sea-level rise, referenced earlier, and similar
to conclusions from Frederikse et al. (2021).

Streamflow explains a substantial portion of the sea-level variation at Buenos Aires, and to a lesser extent Montevideo, and
largely accounts for the apparent faster-than-global rate of regional sea-level rise (Figures 6, 7). To quantify the influence of
streamflow on sea level, I evaluate a multiple linear regression model at each virtual station, where sea level is the dependent
variable and streamflow, time, and unity are the independent variables.[1] The streamflow regressor explains $59 \pm 17\%$, $28 \pm 21\%$,
and $-6 \pm 9\%$ of the sea-level variance at Buenos Aires, Montevideo, and Mar del Plata, respectively (Figure 6). This suggests
that streamflow has more of an influence on sea level closer to the mouths of the Río Paraná and Río Uruguay, generally.
Regression coefficients between streamflow and sea level for Buenos Aires, Montevideo, and Mar del Plata are $(7.3 \pm 1.8) \times$
$10^{-6}$, $(4.7 \pm 2.6) \times 10^{-6}$, and $(-1.1 \pm 1.6) \times 10^{-6}$ m m$^{-3}$ s, respectively (Figure 7). This structure shows that sea level is more
sensitive to streamflow closer the mouths of the rivers. Finally, linear trends computed from the virtual-station time series from
this regression model are $0.75 \pm 0.34$, $0.56 \pm 0.58$, and $1.11 \pm 0.37$ mm yr$^{-1}$ at Buenos Aires, Montevideo, and Mar del Plata,
respectively (Figure 7). Compared to trends reported in the last paragraph, this implies that streamflow effected sea-level rates
of $0.71 \pm 0.35$, $0.48 \pm 0.38$, and $-0.11 \pm 0.17$ mm yr$^{-1}$ at the respective virtual stations (Figure 7). Averaging the streamflow-
corrected sea-level trends, and adjusting for the background geologic rate, I obtain a mean rate of $1.34 \pm 0.40$ mm yr$^{-1}$, which
is more in line with recent global-mean sea-level trends for the past century from Hay et al. (2015), Dangendorf et al. (2017),
and Frederikse et al. (2020).

---

[1]To establish the robustness of the results, I also considered alternative models and analysis approaches. First, I evaluated the same model but using ridge
regression. This was meant to account for collinearity between predictors (e.g., the linear trend in streamflow). Results obtained for a wide range of ridge-
parameter values were essentially identical to the results found from ordinary least squares discussed in the main text (not shown). From this, I concluded that
the model is well posed, and that collinearity between streamflow and time does not pose a serious issue. Second, I evaluated the same model using ordinary
least squares but considering sea-level and streamflow data with ENSO effects removed prior to analysis. I removed ENSO effects by regressing the quantity
of interest against the Niño 3.4 Index and its Hilbert transform to capture arbitrary phase relationships between quantities. If river effects on sea level were
restricted to ENSO events, then results from this analysis should give no meaningful relationship between sea level and streamflow. However, in this analysis,
I found very similar regression coefficients between sea level and streamflow [$(6.9 \pm 1.7) \times 10^{-6}$ at Buenos Aires; $(3.9 \pm 2.6) \times 10^{-6}$ at Montevideo;
$(-1.3 \pm 1.8) \times 10^{-6}$ at Mar del Plata] and sea-level variance explained by streamflow ($55 \pm 18\%$ at Buenos Aires; $21 \pm 19\%$ at Montevideo; $-7 \pm 10\%$ at
Mar del Plata) as previously when I did not remove ENSO effects prior to analysis. From this, I concluded that river effects on sea level in the Río de la Plata
are not restricted to ENSO events, which have been the focus of past studies cited above, but are rather more general.



## 4 Interpretation

Findings in the preceding section are based on correlation and regression analysis. They do not necessarily demonstrate that streamflow and coastal sea level are causally connected. To provide physical interpretation and establish causality, I develop simple theories for the relationship between streamflow and coastal sea level based on ocean dynamics in Sections 4.a and 4.b, and compare model predictions to observational results in section 4.c.

### 4.1 Theory for Buenos Aires

Around Buenos Aires and Palermo, the Río de la Plata is relatively shallow, narrow, and fresh (Guerrero et al., 1997). To model sea level in this region, I use the following conservation laws

$$u_x + v_y + w_z = 0, \tag{1}$$

$$p_z = -\rho_f g, \tag{2}$$

$$0 = -\frac{1}{\rho_f} p_x + \nu u_{zz}. \tag{3}$$

Here $u$, $v$, and $w$ are velocities in along-estuary ($x$), across-estuary ($y$), and vertical ($z$) directions, respectively, $p$ is hydrostatic pressure, $\rho_f$ is a reference fresh water density, $g$ is acceleration due to gravity, $\nu$ is kinematic viscosity, and $x$, $y$, and $z$ subscripts are spatial derivatives. Equations (1) and (2) are familiar forms of the continuity equation and hydrostatic balance (Gill, 1982). Equation (3) specifies along-estuary momentum conservation in terms of a balance between pressure gradient and viscous forces; it omits the time tendency given the long periods under consideration; it also neglects nonlinear advection and Coriolis acceleration under the assumptions of small Reynolds number and large Ekman number, which are reasonable given the spatial scales of the problem.

Integrating Equation (1) over the depth $H(x)$ and width $W(x)$ of the estuary, applying kinematic boundary conditions at the bottom and along the sides, and ignoring the time tendency gives

$$(\langle \overline{u} \rangle W H)_x = 0, \tag{4}$$

where overbar and bracket are depth and across-estuary average, respectively. Integrating Equation (2) vertically, substituting into Equation (3), and averaging over depth and width yields

$$0 = -g \langle \zeta \rangle_x - \frac{C_d U}{H} \langle \overline{u} \rangle, \tag{5}$$

where $\zeta$ is ocean-dynamic sea level, $C_d$ is a drag coefficient, and $U$ is a reference velocity scale. To obtain Equation (5), I assumed that the $\zeta$ slope across the estuary is linear, and that

$$\nu u_z = C_d U \overline{u}, \tag{6}$$

along the bottom. To solve Equations (4) and (5) for $\langle \zeta \rangle$, I specify that along-estuary transport equals the streamflow $q$ at the origin

$$\langle \overline{u} \rangle W H = q \text{ at } x = 0, \tag{7}$$





and that $\langle \zeta \rangle$ vanishes far from the source

$$\lim_{x \to \infty} \langle \zeta \rangle = 0. \tag{8}$$

Combining Equations (4) and (7), substituting for $\langle \overline{u} \rangle$ in Equation (5), integrating along the estuary from $x$ to $\infty$, and applying the boundary condition from Equation (8) gives

$$\langle \zeta \rangle = \frac{C_d U q}{g} \int\limits_x^\infty \frac{1}{H^2 W} dx', \tag{9}$$

for arbitrary depth and width profiles. For an estuary with exponential width and depth (Figure 2)

$$W = W_0 \exp\left(x/L_W\right), \tag{10}$$
$$H = H_0 \exp\left(x/L_H\right), \tag{11}$$

where $W_0$ and $H_0$ are initial values and $L_W$ and $L_H$ are length scales, the solution to Equation (9) is

$$\langle \zeta \rangle = \left(\frac{2}{L_H} + \frac{1}{L_W}\right)^{-1} \frac{C_d U q}{g H^2 W}. \tag{12}$$

The $\langle \zeta \rangle$ response is linear in $q$, and controlled by friction and the geometry of the estuary; it is larger for stronger friction $C_d U$, narrower initial width $W_0$, shallower initial depth $H_0$, longer width and depth scales $L_W$ and $L_H$, and decays rapidly with distance from the origin (Figure 8).

Regression coefficients computed between sea-level and streamflow data (Figure 7) can be understood as approximate ob-
servational estimates of the derivative of the former with respect to the latter. From Equation (12), it follows that

$$\langle \zeta \rangle_q = \left(\frac{2}{L_H} + \frac{1}{L_W}\right)^{-1} \frac{C_d U}{g H^2 W}. \tag{13}$$

Below, I evaluate Equation (13) numerically and compare the values to the empirically determined regression coefficients to test whether the theory is consistent with the observations.

### 4.2 Theory for Montevideo

The solution for Buenos Aires [Equation (12)] is not applicable to Montevideo. The estuary becomes wider, deeper, and more saline by this point (Guerrero et al., 1997; Figures 1, 2), hence stratification and rotation effects cannot be neglected as they were previously. I develop a theory for the $\zeta$ response at Montevideo building on past studies of bottom-advected (slope-controlled) plumes (Chapman and Lentz, 1994; Lentz and Helfrich, 2002; Yankovsky and Chapman, 1997). I take $x$, $y$, and $z$ to be the offshore, alongshore, and vertical coordinates, respectively. As a mental model, I envision a narrow alongshore jet
over a sloping bottom $H(x)$ in thermal-wind balance with a sharp density front some distance $x_p$ offshore (e.g., Lentz and Helfrich, 2002, Figure 3). I imagine the jet transport includes both the fresh river water and salty ocean water brought into the





plume by turbulent mixing. These features are represented by the following governing equations

$$fv = \frac{1}{\rho_0}p_x, \tag{14}$$

$$p_z = -\rho g, \tag{15}$$

$$Q = q + E, \tag{16}$$

$$\frac{Q}{H}\int_{-H}^{0}\rho(x,z)dz = q\rho_f + E\rho_0, \tag{17}$$

where $f = 2\Omega\sin\phi$ is the Coriolis frequency for Earth rotation rate $\Omega$ and latitude $\phi$, $\rho_0$ is an ambient ocean density, $Q$ is volume transport of the vertically sheared geostrophic jet, and $E$ is entrainment flux. Equations (14) and (15) are geostrophic and hydrostatic balances, respectively. Equation (16) is a form of the continuity equation, which states that volume is conserved within the jet. Density conservation in Equation (17) is equivalent to steady state heat and salt conservation for a linear equation of state.[2] Boundary conditions are that alongshore velocity vanishes everywhere along the bottom, and that velocity shear is zero at the foot of the front (Chapman and Lentz, 1994; Lentz and Helfrich, 2002; Yankovsky and Chapman, 1997),

$$v = 0 \text{ at } z = -H(x),\ \forall x \tag{18}$$

$$v_z = 0 \text{ at } x = x_p,\ z = -H(x_p) \doteq -H_p. \tag{19}$$

A solution to Equations (14)–(19) is obtained by giving a functional form to the density field. I picture an infinitely narrow front, with ambient ocean density everywhere offshore, and a mixture of fresh river water and salty ocean water onshore of the front, which I model as (Figures 9a, 9b)

$$\rho(x,z) = \rho_0 + \frac{\rho'}{H_p}(z+H_p)\left[\mathcal{H}(x-x_p)-1\right], \tag{20}$$

where $\rho'$ is a density increment and $\mathcal{H}$ is the Heaviside step function. The alongshore velocity field in thermal-wind balance with this density structure, obtained by cross differentiating Equations (14) and (15) and then integrating vertically subject to the boundary conditions, is

$$v(x,z) = -\frac{g\rho'}{2\rho_0 f H_p}(z+H_p)^2\delta(x-x_p), \tag{21}$$

where $\delta$ is the Dirac delta (Figure 9c).

To obtain the sea-level solution corresponding to Equation (21), I integrate geostrophic balance at the surface

$$fv = g\zeta_x, \tag{22}$$

over all offshore locations, which gives

$$\zeta = \frac{\rho' H_p}{2\rho_0}\left[1-\mathcal{H}(x-x_p)\right]. \tag{23}$$

---

[2]Strictly speaking, since its left-hand side is equivalent to $\int \overline{\rho}v dz$, where overbar is again vertical average, Equation (17) is an approximate form of density conservation. Exact density conservation would require the left-hand side to equal $\int \rho v dz$. However, assuming the density and velocity profiles given in Equations (20) and (21), it can be shown that the omitted term $\int (\rho-\overline{\rho})v dz$ is a factor of $\sim \rho'/\rho_0 \approx 10^{-2}$–$10^{-3}$ smaller than $\int \overline{\rho}v dz$, meaning that the approximate nature of Equation (17) is sufficiently accurate for present purposes, and the equal sign is appropriate.



That is, $\zeta$ takes on a constant value of $\rho' H_p / 2\rho_0$ onshore of the front, experiences a step change at the front, and vanishes offshore of the front. The $\zeta$ solution can be written more explicitly in terms of streamflow $q$ and river and ocean densities $\rho_f$

and $\rho_0$ as follows. First, I express $Q$ in terms of $q$ and density. Given Equation (20), the vertically averaged density within the front is

$$\frac{1}{H} \int_{-H}^{0} \rho(x_p, z) dz = \rho_0 - \frac{\rho'}{4}, \tag{24}$$

which, substituting into Equation (17) and combining with Equation (16) to eliminate $E$, implies

$$Q = \frac{4q(\rho_0 - \rho_f)}{\rho'}, \tag{25}$$

which is analogous to a form of Knudsen's hydrographical theorem (Dyer, 1997). Second, I solve for $H_p$ in terms of $Q$ and density. Integrating both sides of Equation (21) over all depths and offshore locations and rearranging gives

$$Q = -\frac{g\rho' H_p^2}{6\rho_0 f}, \tag{26}$$

or, after rearranging and solving for $H_p$ (and recalling that $f < 0$ in the Southern Hemisphere),

$$H_p = \left(-\frac{6Qf\rho_0}{g\rho'}\right)^{1/2}. \tag{27}$$

Finally, I substitute Equation (25) for $Q$ in Equation (27), insert the resulting expression for $H_p$ in Equation (23), and cancel common terms to give

$$\zeta = \left[-\frac{6fq(\rho_0 - \rho_f)}{\rho_0 g}\right]^{1/2} [1 - \mathcal{H}(x - x_p)]. \tag{28}$$

The $\zeta$ response is nonlinear in $q$, and controlled by stratification and rotation; it is larger for higher latitude, stronger streamflow, and sharper density contrast (Figures 9d–9f). While there is no alongshore dependence in Equation (28), it assumes that the

245 location of interest is downstream in the far field of the river mouth. Given Equation (28), the derivative of $\zeta$ with respect to $q$, which can be evaluated numerically and compared to regression coefficients from observations, is

$$\zeta_q = \left[-\frac{3f(\rho_0 - \rho_f)}{2\rho_0 gq}\right]^{1/2} [1 - \mathcal{H}(x - x_p)]. \tag{29}$$

### 4.3 Model-data comparison

To test whether empirical results from Section 3 are consistent with theories developed in Sections 4.a and 4.b, I evaluate

Equation (13) for Buenos Aires and (29) for Montevideo using parameter values in Table 4, and then compare the predictions to the observed values (Figure 7). Equation (13) gives a theoretical regression coefficient between streamflow and sea level for Buenos Aires of $(7.0 \pm 4.0) \times 10^{-6}$ m m$^{-3}$ s, where the error bar reflects uncertainties on the parameter values (Table 4). Multiplying this coefficient by the longterm trend in streamflow estimated earlier ($96 \pm 37$ m$^3$ s$^{-1}$ yr$^{-1}$), I obtain an expected sea-level trend at Buenos Aires due to streamflow of $0.68 \pm 0.47$ mm yr$^{-1}$. These theoretical estimates agree with



the coefficient of $(7.3 \pm 1.8) \times 10^{-6}$ m m$^{-3}$ s and the streamflow-driven sea-level trend of $0.71 \pm 0.35$ mm yr$^{-1}$ found earlier from regression analysis of observed streamflow and sea level at Buenos Aires (Figure 7). Following the same approach, and evaluating Equation (29), I find a theoretical regression coefficient of $(4.0 \pm 0.1) \times 10^{-6}$ m m$^{-3}$ s and an anticipated sea-level trend forced by streamflow of $0.41 \pm 0.19$ mm yr$^{-1}$ for Montevideo. Again, these values from first principles are consistent with the regression coefficient of $(4.8 \pm 2.7) \times 10^{-6}$ m m$^{-3}$ s and the streamflow-induced sea-level trend of $0.48 \pm 0.38$ mm

260 yr$^{-1}$ found from the observational data (Figure 7). The consistency between theory and observation suggests that the statistical connections found earlier between measured streamflow and sea level at Buenos Aires and Montevideo identify cause-and-effect relationships, which are consistent with the physics prescribed above.

The lack of a significant relation between streamflow and sea level in Mar del Plata in the data (Figures 6, 7) is also consistent with the theories developed in Sections 4.a and 4.b. The response described by Equation (12) imagines a rapid decay away

from the rivers. Indeed, given its strong exponential dependence, the sea-level response predicted by this theory is vanishingly small at Mar del Plata (Figures 1, 8). The response described by Equation (28) envisions coastal sea level coupled to a buoyant longshore current in the sense of coastal waves: counter-clockwise along the Uruguay coast and then equatorward along the Brazil coast (Piola et al., 2005). In other words, given this mechanism, Mar del Plata is not downstream of the Río de la Plata, hence no signals are communicated between the two locations according to these physics.

**5 Conclusions**

The Río de la Plata estuary in South America features the longest tide-gauge records in the South Atlantic Ocean (Figures 1, 2). However, the causes of longterm relative sea-level changes in this region have not been firmly established. I interrogated data (Figures 3–5) and developed theories (Figures 8, 9) to argue for cause-and-effect relationships between low-frequency streamflow and sea-level changes in the Río de la Plata over 1931–2014 (Figures 6, 7). Streamflow forcing explained one

half of the sea-level variance on interannual and longer time scales observed at Buenos Aires and one-quarter of the sea-level variance at Montevideo over the study period, generally. Specifically, a trend in streamflow of $\sim 100$ m$^3$ s$^{-1}$ yr$^{-1}$ during the past century caused sea level to rise at rates of $\sim 0.7$ mm yr$^{-1}$ at Buenos Aires and $\sim 0.5$ mm yr$^{-1}$ at Montevideo. These findings advance understanding of local, regional, and global sea-level changes; clarify basic sea-level physics; inform future projections of coastal sea-level change as well as the interpretation of satellite data and proxy reconstructions; and highlight

future research directions.

This paper complements past tide-gauge studies on mean sea-level changes in the Río de la Plata on interannual to centennial time scales (e.g., Aubrey et al., 1988; Brandani et al., 1985; D'Onofrio et al., 2008; Dennis et al., 1995; Douglas, 1997, 2001, 2008; Emery and Aubrey, 1991; Fiore et al., 2009; Frederikse et al., 2021; Isla, 2008; Lanfredi et al., 1998; Meccia et al., 2009; Melini et al., 2004; Papadopoulous and Tsimplis, 2006; Pousa et al., 2007; Raicich, 2008; Santamaria-Aguilar et al.,

2017; Thompson et al., 2016; Verocai et al., 2016). Previous authors establish that streamflow and sea level in the Río de la Plata covary on interannual time scales during ENSO events, but they do not identify the causal mechanisms responsible for the observed statistical correlations, nor do they consider how these two variables correspond more generally on longer





time scales. My paper builds on their foundation by showing that river effects on sea level are not restricted to ENSO events in particular, but are also apparent more generally at multidecadal and centennial periods, and by identifying ocean-dynamic
mechanisms that mediate the relationship between streamflow and sea level. These results corroborate the hypothesis due to Douglas (2001) that interannual sea-level variation at Buenos Aires over the 1982–1983 El Niño can be understood in terms of ocean-dynamic processes, but they do not necessarily falsify suggestions that contemporary gravitational, rotational, and deformational effects also played a role (Isla, 2008; Thompson et al., 2016). Likewise, while they suggest that streamflow changes contributed importantly to longterm sea-level rise observed at Buenos Aires and Montevideo, these results do not rule
out the possibility that other geophysical processes also effected regional sea-level trends (Melini et al., 2004; Aubrey et al., 1988).

My results have implications for twentieth-century global sea-level reconstructions and budgets (e.g., Church and White, 2011; Dangendorf et al., 2017; Frederikse et al., 2018, 2020, 2021; Hamlington and Thompson, 2015; Hay et al., 2015; Jevrejeva et al., 2014; Natarov et al., 2017; Ray and Douglas, 2011; Thompson and Merrifield, 2014; Thompson et al., 2016). The
streamflow-driven sea-level effects highlighted here are local to regional in scale; they do not contribute meaningfully to sea-level changes on basin or global scales. Hence, such river effects on tide gauges in the Río de la Plata should be removed prior to analysis if the data are used in large-scale circulation and climate studies, lest this local or regional "noise" alias onto the basin or global "signal" of interest (e.g., Papadopoulous and Tsimplis, 2006; Thompson et al., 2016). Given the heavy weight placed on tide gauges from the Río de la Plata, streamflow-driven ocean dynamics could contribute to the lack of sea-level-
budget closure and faster-than-global trends across the South Atlantic during the twentieth century found by Frederikse et al. (2018, 2021). Since tide-gauge records in and around the Río de la Plata are the main (if not sole) data constraint in the South Atlantic prior to 1950 in twentieth-century global-mean sea-level reconstructions (Figure 1b in Hamlington and Thompson, 2015; Figure S1a in Dangendorf et al., 2017), it would be informative to estimate twentieth-century global-mean sea-level rise from tide-gauge records adjusted for river effects, which are typically not considered in global budgets and reconstructions.

Theories developed here [Equations (13) and (29)] clarify relationships between streamflow and coastal sea level, the physics of which have not been well understood (Durand et al., 2019). Piecuch et al. (2018a) formulate a theory for the far-field coastal sea-level response to buoyant river discharge in the limit of a pure surface-advected plume [their Equations (5) and (6)]. This study improves upon their work in two ways. First, I developed a barotropic theory for the sea-level response within an estuary [Equation (13)], where frictional effects and the shape of coastlines and bathymetry are important. Second, I formulated a far-
field theory for the coastal sea-level adjustment in the alternative limit of a purely bottom-advected (or slope-controlled) plume [Equation (29)], which is more suited to the problem at hand. These new theories allow the relationship between sea level and river discharge to be studied in a wider range of settings. In a future study, I plan to develop a more general far-field theory for the buoyancy-driven sea-level response to an intermediate buoyant plume that falls between the extremes of a surface-advected plume and a bottom-advected plume (Yankovsky and Chapman, 1997; Lentz and Helfrich, 2002).

I demonstrated that the sea-level response to buoyant coastal discharge can depend sensitively on density gradients over short scales and the geometry of coastlines and bathymetry. With some exceptions (Haarsma et al., 2016), the current generation of coupled models used for climate projections are too coarsely resolved to represent such features (Holt et al., 2017). Theories





developed here may be helpful in this regard. Equations (13) and (29) may be instructive for obtaining basic scales and magnitudes of future coastal sea-level changes due to streamflow, assuming that the details of coastlines and bathymetry are

325 known, and given projected changes in continental freshwater runoff into the coastal ocean.

Due to my focus on longterm trends, I interrogated sea-level records from tide gauges. However, streamflow-driven sea-level changes are also apparent in data from other observing systems, including satellite altimetry. Comparing annual streamflow and sea-surface-height anomaly from along-track altimetry over 1993–2014 (Birol et al., 2017), I observe a region of significant correlation between the two variables extending broadly over the Uruguay coast from Montevideo past La Paloma towards

Brazil, and onshore of the $\sim 100$-m isobath (Figure 10a; cf. Figure 1). The shape of the region mirrors the structure of low-salinity water near the mouth of the estuary (e.g., Piola et al., 2005). Regression coefficients obtained between Río de la Plata streamflow and sea-surface-height anomaly are consistent with theoretical expectations: more upstream in the estuary, values are $\lesssim 1 \times 10^{-5}$ m m$^{-3}$ s, similar to predictions from barotropic theory developed in Section 4.a [Equation (13)], whereas values downstream in the far field are $\sim 4 \times 10^{-6}$ m m$^{-3}$ s, consistent with values anticipated from the baroclinic theory from

Section 4.b [Equation (29)] (Figure 10b; cf. Figure 7). The offshore extent of the region of significant correlation between streamflow and sea-surface height also corroborates basic theoretical expectations: for strong slope control and large river discharge, the offshore and vertical scales of a buoyant coastal plume are expected to be $\sim 100$ km and $\sim 100$ m, respectively (e.g., Yankovsky and Chapman, 1997).

Findings here may have implications for proxy reconstructions of late-Holocene sea level from natural archives, which have

340 temporal resolution of decades to centuries (e.g., Kemp et al., 2009; Khan et al., 2019). Whereas past studies reason river effects contribute to sea-level variability on interannual and shorter time scales (e.g., Durand et al., 2019; Woodworth et al., 2019), I showed that streamflow changes can be an important driver of sea-level changes over multidecadal and longer periods. This result has (at least) two important implications for proxy reconstructions. First, it implies that river effects may be important to consider when interpreting proxy sea-level reconstructions from large rivers or estuaries (e.g., Gerlach et al., 2017; Kemp et al.,

2018). Second, it suggests that proxy sea-level reconstructions produced from strategic locations may inform past changes in streamflow, and thus complement estimates from more traditional archives like tree rings (e.g., Margolis et al., 2011; Devineni et al., 2013).

Other major rivers including the Mississippi, Yenisey, and Lena have also undergone significant streamflow trends in the past century (e.g., Dai, 2016; Dai and Trenberth, 2002; Dai et al., 2009). However, the effect of these historical changes in

streamflow on longterm sea-level change has not been considered. Future studies should take advantage of the growing number of available runoff and streamflow datasets (e.g., Do et al., 2018; Gudmundsson et al., 2018; Tsujino et al., 2018) to test the analytical models developed here and observationally constrain river effects on historical sea-level rise more globally, which could inform studies of ocean circulation and climate change.

*Data availability.* All data used here are publicly available. Tide-gauge data are available through the Permanent Service for Mean Sea

Level (https://www.psmsl.org/). Stream-gauge data are available through the Global Streamflow Indices and Metadata Archive



(https://doi.pangaea.de/10.1594/PANGAEA.887470). Proxy reconstructions are taken from the appendix of Milne et al. (2005). Bathymetry data are available through the GEBCO Compilation Group 2021 (https://www.gebco.net/). Altimetry data are from the Center for Topographic studies of the Ocean and Hydrosphere (http://ctoh.legos.obs-mip.fr/).

## Appendix A: Bayesian hierarchical model

I apply Bayesian linear regression to proxy reconstructions from Milne et al. (2005) to quantify late-Holocene rates of sea-level change. Bayesian linear regression is chosen over more traditional approaches like least squares or maximum likelihood because Bayesian methods provide a more transparent means for incorporating data errors into the formal uncertainty quantification. I design the Bayesian hierarchical model following similar algorithms developed in past studies (Ashe et al., 2019; Cahill et al., 2015, 2016; Walker et al., 2020). The model used here is essentially the time component of the spacetime model

from Piecuch et al. (2018b). While I give a brief description for sake of completeness, readers are referred to Piecuch et al. (2018b) for a more detailed presentation.

Temporal Bayesian hierarchical models comprise three levels: a process level that prescribes the temporal evolution of the sea-level process; a data level that codifies the relationship between the uncertain proxy reconstructions and the sea-level process; and a parameter level where prior constraints are specified.

For the process level, I model sea level $\boldsymbol{y} = [y_1, y_2, \ldots, y_n]^\mathsf{T}$ as a linear function of time $\boldsymbol{x} = [x_1, x_2, \ldots, x_n]^\mathsf{T}$ according to

$$y_k \sim \mathcal{N}\left(\alpha x_k + \beta, \gamma^2\right), \; k \in [1, n], \tag{A1}$$

where $\sim$ means "is distributed as," $\mathcal{N}(a, b^2)$ is the normal distribution with mean $a$ and variance $b^2$, and $\alpha$, $\beta$, and $\gamma^2$ are uncertain slope, intercept, and residual variance parameters, respectively. For the data level, I represent the proxy reconstructions of relative sea level $\boldsymbol{z} = [z_1, z_2, \ldots, z_n]^\mathsf{T}$ and age $\boldsymbol{w} = [w_1, w_2, \ldots, w_n]^\mathsf{T}$ as noisy versions of the respective processes, *viz.*,

$$z_k \;\sim\; \mathcal{N}\left(y_k, \delta_k^2\right), \tag{A2}$$
$$w_k \;\sim\; \mathcal{N}\left(x_k, \epsilon_k^2\right), \tag{A3}$$

where $\delta_k^2$ and $\epsilon_k^2$ are the data error variances, which are provided (Table 3). To close the model, I assume normal priors for $\alpha$ and $\beta$, and an inverse-gamma prior for $\gamma^2$,

$$\alpha \;\sim\; \mathcal{N}\left(\tilde{\mu}, \tilde{\kappa}^2\right), \tag{A4}$$
$$\beta \;\sim\; \mathcal{N}\left(\tilde{\eta}, \tilde{\sigma}^2\right), \tag{A5}$$
$$\gamma^2 \;\sim\; \mathcal{G}^{-1}\left(\tilde{\xi}, \tilde{\chi}\right), \tag{A6}$$

where tildes identify fixed hyperparameters (see below for numerical values).

Given Bayes' rule and the model equations, I assume the posterior distribution is

$$p\left(\boldsymbol{y}, \boldsymbol{x}, \alpha, \beta, \gamma^2 | \boldsymbol{z}, \boldsymbol{w}\right) \propto p(\alpha)\, p(\beta)\, p\left(\gamma^2\right) \prod_{k=1}^{n} \left[p\left(z_k | y_k\right) p\left(w_k | x_k\right) p\left(y_k | x_k, \alpha, \beta, \gamma^2\right)\right], \tag{A7}$$



where $p$ is probability, $|$ is conditionality, and $\propto$ is proportional to. To evaluate the posterior, I use a Gibbs sampler (Gelman et al., 2013), evaluating the full posteriors (Wikle and Berliner, 2007)

$$\alpha| \cdot \ \sim \ \mathcal{N}\left(\left[\tilde{\kappa}^{-2}+\gamma^{-2}\sum_{k=1}^{n}x_k^2\right]^{-1}\left[\tilde{\kappa}^{-2}\tilde{\mu}+\gamma^{-2}\sum_{k=1}^{n}x_k\left\{y_k-\beta\right\}\right],\left[\tilde{\kappa}^{-2}+\gamma^{-2}\sum_{k=1}^{n}x_k^2\right]^{-1}\right),\tag{A8}$$

$$\beta| \cdot \ \sim \ \mathcal{N}\left([\tilde{\sigma}^{-2}+n\gamma^{-2}]^{-1}\left[\tilde{\sigma}^{-2}\tilde{\eta}+\gamma^{-2}\sum_{k=1}^{n}\left\{y_k-\alpha x_k\right\}\right],[\tilde{\sigma}^{-2}+n\gamma^{-2}]^{-1}\right),\tag{A9}$$

$$\gamma^2| \cdot \ \sim \ \mathcal{G}^{-1}\left(\tilde{\xi}+\frac{n}{2},\tilde{\chi}+\frac{1}{2}\sum_{k=1}^{n}[y_k-\alpha x_k-\beta]^2\right),\tag{A10}$$

$$y_k| \cdot \ \sim \ \mathcal{N}\left(\left[\delta_k^{-2}+\gamma^{-2}\right]^{-1}\left[\delta_k^{-2}z_k+\gamma^{-2}\left\{\alpha x_k+\beta\right\}\right],\left[\delta_k^{-2}+\gamma^{-2}\right]^{-1}\right),\tag{A11}$$

$$x_k| \cdot \ \sim \ \mathcal{N}\left(\left[\epsilon_k^{-2}+\alpha^2\gamma^{-2}\right]^{-1}\left[\epsilon_k^{-2}w_k+\gamma^{-2}\alpha\left\{y_k-\beta\right\}\right],\left[\epsilon_k^{-2}+\alpha^2\gamma^{-2}\right]^{-1}\right),\tag{A12}$$

where $|\cdot$ is conditionality on all other processes, parameters, and data. I set weak, uninformative priors ($\tilde{\mu}=0$ mm yr$^{-1}$, $\tilde{\kappa}^2=0.001$ mm$^2$ yr$^{-2}$, $\tilde{\eta}=0$ m, $\tilde{\sigma}^2=100$ m$^2$, $\tilde{\xi}=0.5$, $\tilde{\chi}=0.02$ m$^2$). I discard 1 000 burn-in draws to eliminate startup transients. I reduce autocorrelation of the samples by keeping only every 10th draw of the subsequent 10 000 iterations of the Gibbs sampler. This gives a 1 000-member ensemble of posterior estimates for $\boldsymbol{y}$, $\boldsymbol{x}$, $\alpha$, $\beta$, and $\gamma^2$. Figure 5 shows summary statistics for the posterior solution of $\alpha x+\beta$ for $x$ from 500 BCE to present.

*Author contributions.* C. G. P. was responsible for all aspects of this study including conceptualization, formal analysis, funding acquisition, investigation, methodology, visualization, and writing.

*Competing interests.* No competing interests are present.

*Acknowledgements.* This work was supported by the National Aeronautics and Space Administration's Sea Level Change Team (grants 80NSSC20K1241, 80NM0018D0004) and the National Science Foundation's Paleo Perspectives on Climate Change program (grant OCE-2002485). Helpful conversations with F. M. Calafat, R. C. Creel, C. W. Hughes, S. Jevrejeva, C. Katsman, A. C. Kemp, S. J. Lentz, K. McKeon, M. Passaro, J. Oelsmann, K. Richter, and A. Wise are acknowledged. This paper is a contribution by the International Space Science Institute (ISSI) Team on "Understanding the Connection Between Coastal Sea Level and Open Ocean Variability Through Space Observations" led by F. M. Calafat and S. Jevrejeva.





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



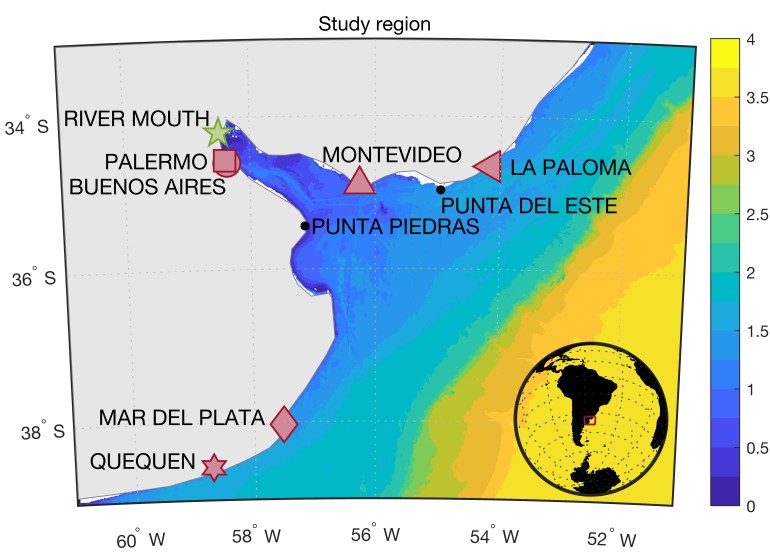

**Figure 1.** Study region. Color shading is $\log_{10}$ of bathymetry (m) from the GEBCO 2021 grid (GEBCO Compilation Group 2021). Red symbols locate tide gauges. Green star is the river mouth, selected as the confluence of the Río Paraná and Río Uruguay near Isla Oyarvide. Black dots identify other locations referenced in the text. Inset shows study area in global context.



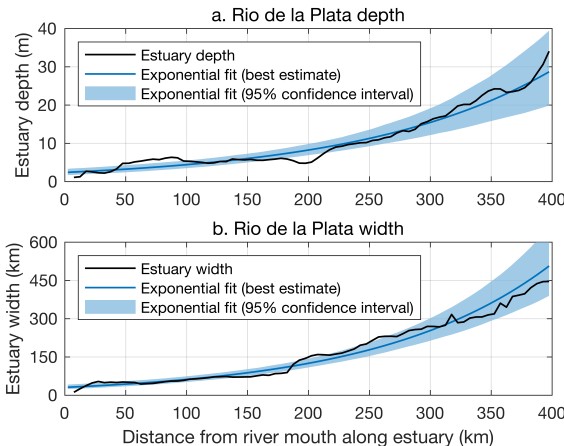

**Figure 2.** Black curves illustrate the **(a.)** average depth and **(b.)** width of the Río de la Plata as a function of distance along the estuary from the river mouth based on the GEBCO 2021 grid (GEBCO Compilation Group 2021). Values are determined by identifying all marine grid cells (depths $< 0$) in successive 5-km increments from the river mouth. The average depth is computed as the arithmetic mean of all grid-cell depths, and the width is defined as the maximum distance between the marine grid cells within the given 5-km increment. Dark blue curves and light blue shading represent best estimates and 95% confidence intervals, respectively, of exponentials fit to the black curves using ordinary least squares. To account for residual autocorrelation, the uncertainties are based on the effective degrees of freedom assuming residuals are described by an order-1 autoregressive model.

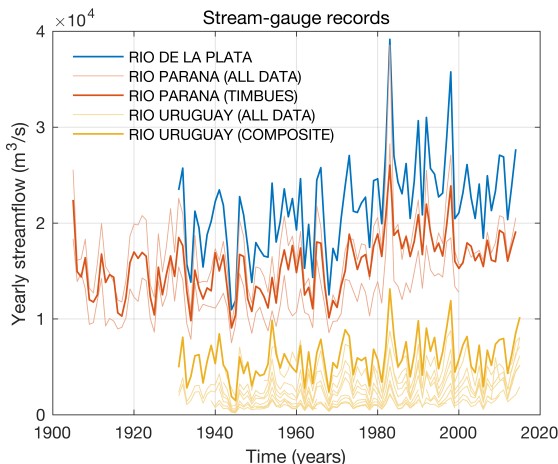

**Figure 3.** Yearly river-gauge streamflow records (Table 1). The thick black Río de la Plata time series is the sum of the thick blue Río Paraná time series from Timbúes and the thick orange composite Río Uruguay time series. Thin time series show data from individual gauges.



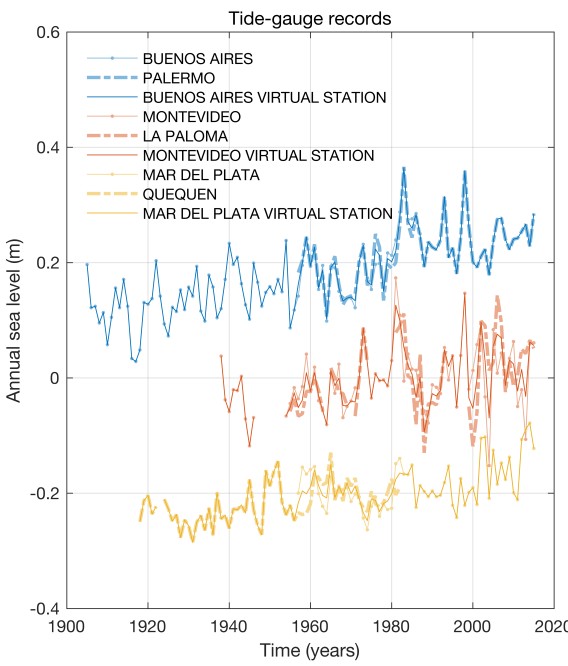

**Figure 4.** Yearly tide-gauge relative sea-level records (Figure 1, Table 2). Virtual-station time series are shown as thick lines and individual tide-gauge records are shown as thin lines. The time series are shifted vertically by an arbitrary amount for ease of visualization.

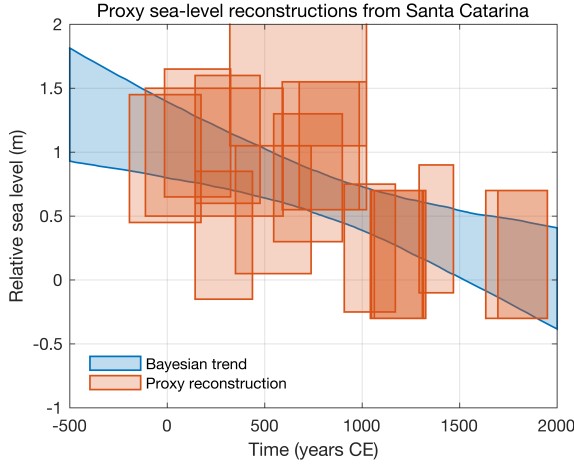

**Figure 5.** Proxy sea-level reconstructions (orange) and Bayesian linear regression (blue). Orange shading identifies best estimates plus and minus twice the standard errors. Blue shading corresponds to 95% posterior credible intervals. The Bayesian model is detailed in the Appendix.

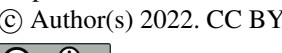


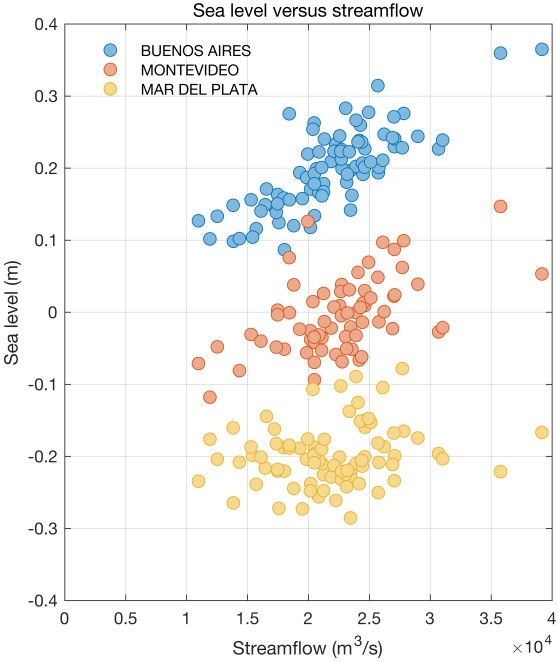

**Figure 6.** Scatter plots comparing yearly average Río de la Plata streamflow (horizontal axes) and relative sea level (vertical axes) at Buenos Aires (blue), Montevideo (orange), and Mar del Plata (yellow). Sea-level values from the different sites are shifted vertically by an arbitrary amount for ease of visualization.

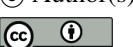



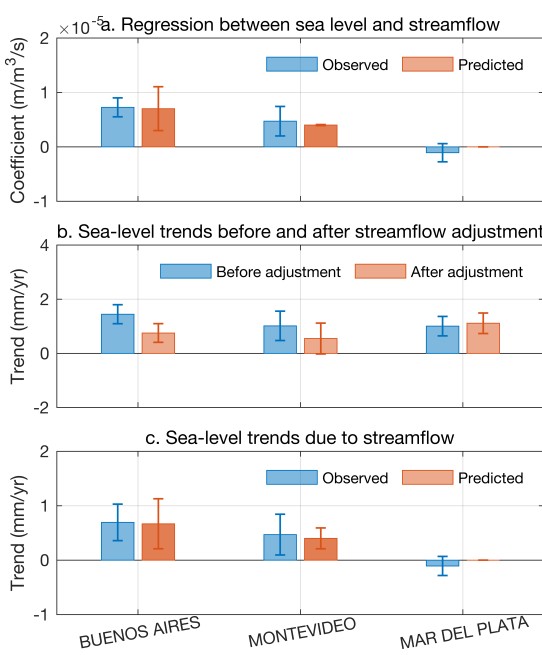

**Figure 7. (a.)** Regression coefficients between sea level and streamflow found empirically from linear regression (blue) and predicted theoretically from ocean dynamics (orange). **(b.)** Trend computed from tide gauges without (blue) and with (orange) adjusting for river effects. **(c.)** Sea-level trend due to streamflow found empirically from linear regression (blue) and predicted theoretically from ocean dynamics given the streamflow trend (orange). To evaluate predicted values at Buenos Aires, I use a value of $x = 65$ km from the source in Equation (13).



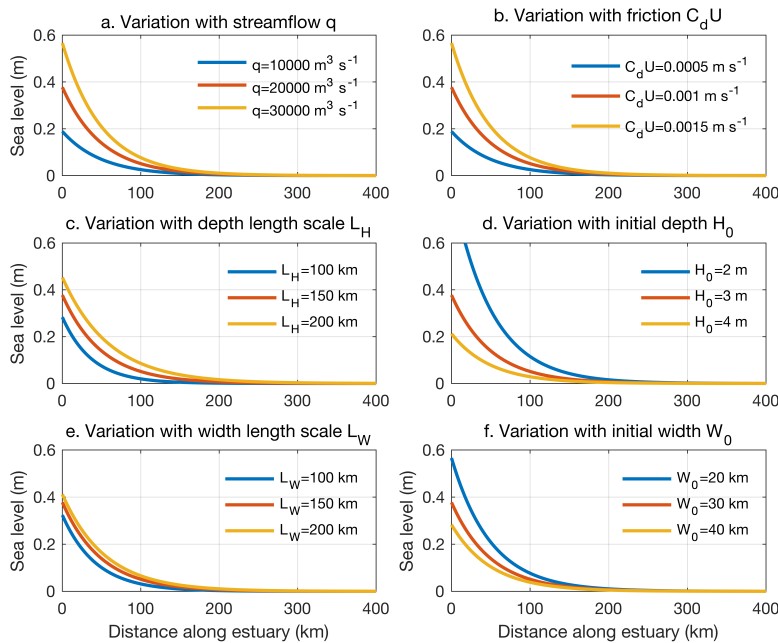

**Figure 8.** Sea-level response $\langle \zeta \rangle$ to streamflow forcing $q$ described by Equation (12) as a function of distance along the estuary away from the mouth of the rivers for different values of **(a.)** streamflow $q$, **(b.)** friction $C_d U$, **(c.)** depth length scale $L_H$, **(d.)** initial depth $H_0$, **(e.)** width length scale $L_W$, and **(f.)** initial width $W_0$. Default values are $q = 2 \times 10^4$ m$^3$ s$^{-1}$, $C_d U = 0.001$ m s$^{-1}$, $L_H = 150$ km, $H_0 = 2$ m, $L_W = 150$ km, and $W_0 = 30$ km.





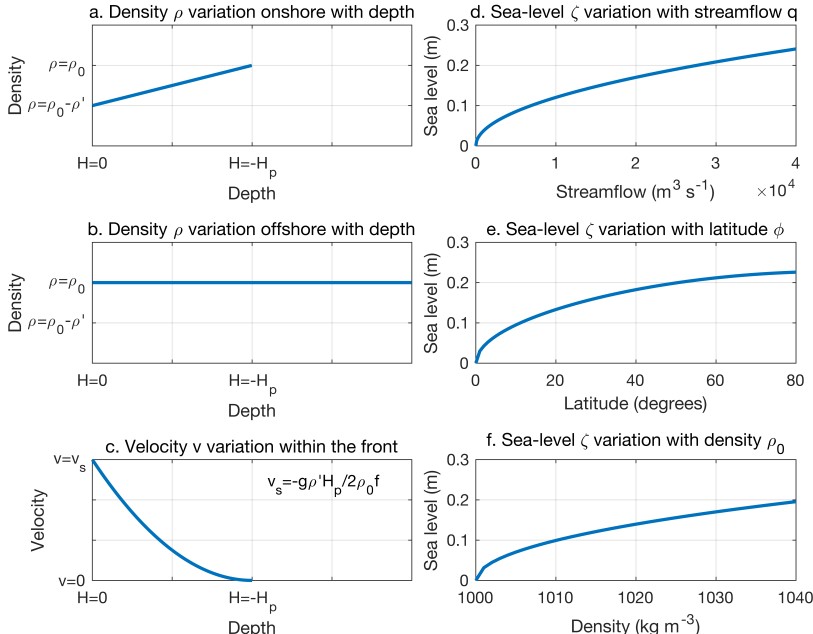

**Figure 9.** Idealized **(a.)** density structure onshore of the front [Equation (20)], **(b.)** density structure offshore of the front [Equation (20)], and **(c.)** velocity structure within the front [Equation (21)] as a function of depth. Sea-level response $\zeta$ described by Equation (28) as a function of **(d.)** streamflow $q$, **(e.)** latitude $\phi$, and **(f.)** ambient ocean density $\rho_0$. Default values: $q = 2 \times 10^4$ m$^3$ s$^{-1}$, $\phi = 35°$, $\rho_0 = 1030$ kg m$^{-3}$.

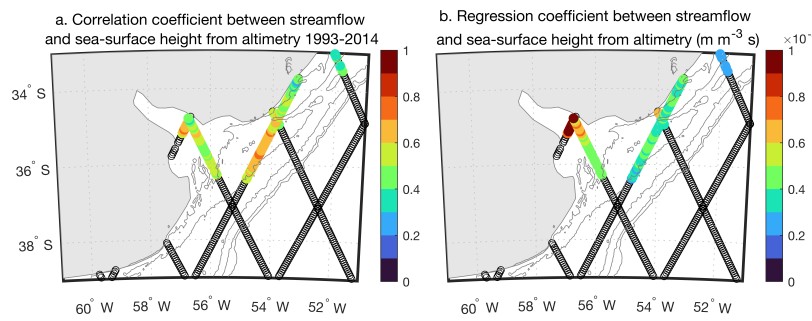

**Figure 10. (a.)** Correlation coefficient and **(b.)** regression coefficient (m m$^{-3}$ s) between annual streamflow in the Río de la Plata (Figure 3) and sea-surface-height anomaly from along-track satellite-altimetry data (Birol et al., 2017) during 1993–2014 over the study regions. Values are only shown where correlation coefficients are positive at the 95% confidence level determined through bootstrapping. Contours identify the 20-, 50-, 100-, 200-, and 500-m isobaths.



| Stream-gauge location | River | GSIM ID | Lon | Lat | Span | Completeness | Area (km$^2$) | Mean flow (m$^3$ s$^{-1}$) |
|---|---|---|---|---|---|---|---|---|
| Posadas | Paraná | AR_0000001 | 55.8 | 27.3 | 1901—2000 | 100% | 975 000 | 12 400 |
| Corrientes | Paraná | AR_0000005 | 58.8 | 27.9 | 1904—2014 | 100% | 1 950 000 | 17 200 |
| Timbúes | Paraná | AR_0000006 | 60.7 | 32.6 | 1905—2014 | 100% | 2 346 000 | 15 600 |
| Marcelino Ramos | Uruguay | BR_0002884 | 51.9 | 27.4 | 1939—1999 | 100% | 40 900 | 910 |
| — | Uruguay | BR_0002887 | 52.3 | 27.2 | 1950—1997 | 92% | 43 900 | 1 020 |
| Passo Caxambu | Uruguay | BR_0002892 | 52.8 | 27.1 | 1940—2010 | 99% | 52 400 | 1 240 |
| — | Uruguay | BR_0002910 | 53.2 | 27.1 | 1941—2016 | 97% | 61 900 | 1 610 |
| Porto Lucena | Uruguay | BR_0002929 | 55.0 | 27.8 | 1931—2007 | 100% | 95 200 | 2 290 |
| Garruchos | Uruguay | BR_0002950 | 55.6 | 28.1 | 1931—2016 | 100% | 116 000 | 2 830 |
| — | Uruguay | BR_0002953 | 56.0 | 28.5 | 2012—2016 | 100% | 120 000 | 3 690 |
| — | Uruguay | BR_0002954 | 56.0 | 28.6 | 1942—2016 | 100% | 125 000 | 3 450 |
| Itaqui | Uruguay | BR_0002956 | 56.5 | 29.1 | 1985—2016 | 47% | 131 000 | 3 590 |
| Paso de los Libres | Uruguay | BR_0002983 | 57.0 | 29.7 | 2012—2016 | 100% | 190 000 | 5 440 |
| Uruguaiana | Uruguay | BR_0002984 | 57.0 | 29.7 | 1942—2016 | 99% | 190 000 | 4 920 |
| Aporte Salto Grande | Uruguay | BR_0002986 | 57.9 | 31.3 | 2012—2016 | 100% | 242 000 | 6 450 |

**Table 1.** GSIM river-gauge records (Do et al., 2018; Gudmundsson et al., 2018; Figure 3). Lon and Lat are degrees west longitude and south latitude, respectively. Completeness is percentage of years during span featuring data. Area is the gauged drainage area. Mean flow is the time-mean streamflow over the record length.

| Tide-gauge location | PSMSL ID | Lon | Lat | Span | Completeness |
|---|---|---|---|---|---|
| Buenos Aires | 157 | 58.37 | 34.60 | 1905–1987 | 100% |
| Palermo | 832 | 58.40 | 34.57 | 1957–2019 | 98% |
| Montevideo | 431 | 56.25 | 34.90 | 1938–2018 | 80% |
| La Paloma | 764 | 54.15 | 34.65 | 1955–2018 | 71% |
| Mar del Plata | 819 | 57.52 | 38.03 | 1957–2019 | 95% |
| Quequén | 223 | 58.70 | 38.58 | 1918–1982 | 99% |

**Table 2.** PSMSL tide-gauge records (Holgate et al., 2013; Figures 1, 4). Lon and Lat are degrees west longitude and south latitude, respectively. Completeness is percentage of years during span that feature data.





| Calendar Age (yr CE) | Age error (yr) | Relative sea level (m) | Sea level error (m) |
|---|---|---|---|
| -10.5 | 92 | 0.95 | 0.25 |
| 155.5 | 85 | 1.15 | 0.25 |
| 241 | 177 | 1 | 0.25 |
| 290 | 74 | 0.35 | 0.25 |
| 309.5 | 83 | 1.1 | 0.25 |
| 544 | 97 | 0.55 | 0.25 |
| 671.5 | 175 | 1.55 | 0.25 |
| 722 | 88 | 0.8 | 0.25 |
| 806 | 108 | 1.05 | 0.25 |
| 831 | 77 | 1.05 | 0.25 |
| 1039 | 66 | 0.25 | 0.25 |
| 1175.5 | 67 | 0.2 | 0.25 |
| 1181.5 | 67 | 0.2 | 0.25 |
| 1194 | 66 | 0.2 | 0.25 |
| 1380 | 44 | 0.4 | 0.25 |
| 1792 | 79 | 0.2 | 0.25 |
| 1823 | 64 | 0.2 | 0.25 |

**Table 3.** Proxy sea-level reconstructions for the past two millennia from Santa Catarina (Milne et al., 2005). Milne et al. (2005) give calendar ages as min-max ranges, which I take to be 95% confidence intervals. I take the center point as the best estimate, and one-quarter of the range as one standard error. I also assume sea-level errors given by Milne et al. (2005) correspond to two standard errors.



| Parameter | Numerical value |
|-----------|-----------------|
| $C_d$ | $2 \times 10^{-3}$ |
| $f$ | $-8.3 \times 10^{-5}$ s$^{-1}$ |
| $g$ | 9.81 m s$^{-2}$ |
| $H_0$ | $2.4 \pm 0.9$ m |
| $L_W$ | $140 \pm 25$ km |
| $L_H$ | $160 \pm 43$ km |
| $q$ | $(2.2 \pm 0.1) \times 10^4$ m$^3$ s$^{-1}$ |
| $\rho_f$ | 1 000 kg m$^{-3}$ |
| $\rho_0$ | 1 030 kg m$^{-3}$ |
| $U$ | $0.4 \pm 0.1$ m s$^{-1}$ |
| $W_0$ | $31 \pm 8.9$ km |

**Table 4.** Parameter values used to evaluate Equations (13) and (29). Values for $C_d$, $f$, $\rho_f$, $\rho_0$, and $g$ are standard. Values for $W_0$, $H_0$, $L_W$, and $L_H$ are based on bathymetry data (Figure 2). I set $U = 0.4 \pm 0.1$ m s$^{-1}$ based on multiplying regional tidal-current amplitudes, on the order of $0.65 \pm 0.15$ m s$^{-1}$ (O'Connor, 1991; Piedra-Cueva and Fossati, 2007), by a factor $2/\pi$, the average amplitude of a sine wave. The $q$ value is the time-mean of the Río de la Plata streamflow time series in Figure 3.