# Peer review of "River effects on sea-level rise in the Río de la Plata Estuary during the past century"

_EGUsphere, 2022_

## Author Comment (AC1)

Response to Reviewer 1
*River effects on sea-level rise in the*
*Río de la Plata during the past century*

by Christopher G. Piecuch

**\*\*\*Reviewer's comments in black\*\*\***
**\*\*\*Author's responses in red\*\*\***

Dear Reviewer 1,

Thank you for your comments on my manuscript. They made the paper stronger, clearer, and more precise. I revised the manuscript based on your reviews, and point-by-point responses follow below. Thank you for your time and consideration.

Best regards,

Christopher G. Piecuch
* * *
**Reviewer 1**—

Review of "River effects on sea-level rise in the Río de la Plata during the past century" by C. Piecuch (OS 2022)

**Summary**: This manuscript investigates the dynamic link between the variability of Rio de la Plata discharge and coastal sea level variability observed in the immediate vicinity of its outlet, primarily focusing on the long timescales. It starts with an in-depth analysis of the few long tide-gauge and streamwflow datasets available there, which reveal increasing trend of both the discharge and the coastal sea level over the past century. Then it proceeds with two idealized modeling frameworks that are derived in order to explain the observed co-existence of discharge and sea level trends, in a causal fashion. The first framework is essentially a one-dimensional barotropic frictional model, where the water-air interfacial slope is balanced by the bottom friction in the along-estuary direction. This framework is applied to the inner estuary, around Buenos Aires. The second framework is a more complex two-dimensional baroclinic framework, where the plume would induce a coastal jet of brackish water, in thermal wind balance with a sea level higher at the coast than further offshore, off the plume offshore edge. It is applied further downstream, around Montevideo. It is concluded that both these modelings results, although based on highly idealized assumptions, stand in very good agreement with the observed relationship between discharge trends and sea level trends, for both regions. Hence it is concluded that the link between Rio de la Plata discharge trend and sea level trend in these two regions is causal in nature.

**General comments**: This study nicely tackles the long-lasting issue of long-term trends of sea level in estuarine ambients. It does so in a very relevant region, home of virtually the sole long-term observational records of the east coast of South America, itself in a very poorly observed basin as far as sea level is concerned: the southern Atlantic. The author makes clever use of the few observational records available, be it of streamflow or of coastal sea level. The approach is sound, and the results are convincing, in that the manuscript consistently backs its findings with statistical analyses. The merit of the study is to end up with a very simple conceptual framework that manages to explain the potentially complex and non-linear dynamics underlying the observed relationship between discharge and coastal sea level. This said, I am somewhat doubtful about the practical strategy of the author when it comes to test the validity of his two idealized frameworks against his observational findings, through the choice of numerical parameters of the required quantities (see my specific comment hereafter). I strongly encourage him to assess the relevance of his idealized models against one (or ideally several) OGCM outputs, typically considering the latest class of eddy-admitting CMIP6 historical runs. This would add considerable strength to the present manuscript. There are very concrete and practical implications of the findings reported here, as regards to the general understanding of the sea level budget and its closure over the southern Atlantic basin, as the observational databases available therein are deeply influenced by the handful of stations analyzed in the present manuscript.

I thank the reviewer for their positive review. Below they will find responses to all their comments.

**Specific comments**:

**R1.CA** l. 90: "To (...) reduce dimensionality": isn't there a more fundamental reason than just reducing dimensionality? Reducing the impact of observational errors, typically?

The revision identifies error reduction as a motivation for averaging the data.

**R1.CB** l. 195: "As a mental model": an actual schematic would help the reader here.

The revision includes a schematic illustration to aid interpretation (Figure 9).

**R1.CC** Table 4: shouldn't $U$ be simply dictated by the geometry of the estuary at $x = 0$, given the value of $q$ observed? It is unclear what relevance the tidal current has in this steady-state model. Is the value chosen for $H_0$ some sort of optimum resulting from a tuning, so as to achieve best consistency of the prediction wrt to the result of the regression analysis based on observations?

The reviewer identifies good questions here and in **R1.CD** and **R1.CE** below. To address these issues, I've now included a new Appendix B, which describes in more detail how and why the numerical values of the various model parameters were selected:

- *To evaluate Equations (13) and (29), numerical values need to be assigned to the various model parameters. Here I detail the rationale behind the*

[Figure]

Figure 9: *Schematic of study region with key model quantities identified. Top shows plan view of region. Bottom shows cross sections at various locations in and around the Río de la Plata. Locations a and b are upstream near Buenos Aires, where the barotropic theory developed in section 4.1 applies. Location c is downstream of Montevideo near La Paloma, where the baroclinic theory developed in section 4.2 applies. In the bottom, yellow $\otimes$ identifies flow into the page, blue indicates fresher, less dense water, and green denotes saltier, more dense water. Other symbols and quantities are as defined in the text of sections 4.1 and 4.2. Illustration by Natalie Renier, WHOI.*

*values tabulated in Table 4.*

*I chose standard reference values for freshwater density $\rho_f$, seawater density $\rho_0$, gravitational acceleration $g$, and the Coriolis parameter at the latitude of the Río de la Plata.*

*Ranges on the initial estuary width $W_0$ and depth $H_0$, and the width and depth scales $L_W$ and $L_H$ from Table 4 correspond to best estimates plus and minus two standard errors on these parameter values as determined by fitting exponentials in the form of Equations (10) and (11) to the bathymetry data in Figure 2 using nonlinear least squares as described in the Figure 2 caption.*

*The q value is the time-mean of the Río de la Plata streamflow time series in Figure 3 plus and minus twice its standard error.*

*Typical values for $C_d$ range from 0.001 to 0.003 (Adcroft et al., 2018, 2019). I selected a middle-of-the-road value of 0.002.*

*The velocity scale U parameterizes the influence of unresolved processes, and is typically selected to represent tidal motions (Adcroft et al., 2019). Regional tidal-current amplitudes are 0.5–0.8 m s$^{-1}$ (O'Connor, 1991; Piedra-Cueva and Fossati, 2007). Multiplying by a factor $2/\pi$, the average amplitude of a sine wave, gives the range of 0.3–0.5 m s$^{-1}$ used here. This is larger than the background mean flow from river discharge, $\langle \overline{u} \rangle = q/H(x)W(x)$, which is 0.12 m s$^{-1}$ at Buenos Aires ($x = 65$ km), using the q, $H_0$, $W_0$, $L_H$, and $L_W$ values in Table 4 discussed earlier.*

**R1.CD** l. 254-256: "These theoretical estimates agree with the coefficient of $(7.3\pm1.8)\times10^{-6}$ m m$^{-3}$ s and the streamflow-driven sea-level trend of $0.71\pm0.35$ mm yr$^{-1}$ found earlier from regression analysis of observed streamflow and sea level at Buenos Aires": indeed, the values do agree very, very well. Hence it is needed here to get a feel of the extent of ad-hoc tuning, implicit in the choice of parameters listed in table 4, so as to ensure this quasi-perfect match.

Please see my response to **R1.CC**.

**R1.CE** l. 257-260: I have the same concern for the results of the Montevideo idealized model.

Please see my response to **R1.CC**.

**R1.CF** l. 322: "Theories developed here may be helpful in this regard": indeed, nowadays there is a whole batch of centennial model outputs that became open for public use, several of which resolve—at least partly—the baroclinic Rossby radius of deformation at the latitude of Rio de la Plata. As part of CMIP6 for instance, multi-centennial historical simulations of the present climate as well as century-long projections have recently become commonly available (see e.g. Held et al 2019 https://doi.org/10.1029/2019MS001829, among many others). If indeed the two idealized dynamical balances proposed in the present study successfully explain the observed relationship between discharge variability and coastal sea level variability in the estuary at long timescales, this relationship should be captured by these long model simulations that have the full physics required to capture these, and much more (in particular that have realistic mixing schemes, and do not impose the idealized frontal structure of the plume density nor its linearly stratified density profile). These OGCMs have, with their $1/4°$ typical resolution in the ocean, the capability to resolve to a fairly large extent the thermal-wind balance invoked in the present idealized framework. I strongly encourage the author to consider at least one of this class of state-of-the-art model historical simulations, and to assess the relationship between Rio de la Plata discharge trend, along-shore equatorward coastal current trend slightly downstream of the outlet, and cross-shore sea level slope trend, at the long timescales of interest here. If the idealized framework presented in the present

manuscript holds there as well, this would add considerable strength to the results reported here (as the conclusion would not depend in any fashion on the potentially subjective choices of parameters listed in Table 4). The encouraging results observed from altimetry call for such an independent assessment of the idealized framework.

I forgo analysis of high-resolution climate-model output because, while it would substantially expand the paper's scope, it would not add value to the manuscript:

1. As the reviewer acknowledges, one of the paper's merits is that it presents a simple mechanistic framework for interpreting the observed relationship between discharge and coastal sea level. Including the kind of multi-model analysis advocated by the reviewer would require substantially expanding the scope and content of the manuscript. Doing so would detract from the clear and focused message, and result in a fundamentally different paper.

2. It's unclear whether current climate models resolve the relevant physics. For example, GFDL CM4.0, referenced by the reviewer, has a horizontal resolution of $\sim 25$ km. This is larger than the first baroclinic Rossby radius of deformation related to Río de la Plata streamflow, which is $\sim 10$ km (cf. Figure S3 in Piecuch et al., 2018). Indeed, Holt et al. (2017) determine that $1/12°$ models only resolve the first baroclinic Rossby radius in $\sim 8\%$ of ocean regions shallower than 500 m, meaning that much higher resolution models are needed to faithfully represent coastal processes and shelf seas. Likewise, the geometry of the estuary also poses a challenge for models like CM4.0. For instance, that model's 2-m vertical spacing near the surface (section 2.1.4 in Adcroft et al. 2019) is coarse compared to the shallower depths at the head of the estuary (Figures 1 and 2 here).

3. Relatedly, global climate models parameterize boundary friction effects. For example, GFDL CM4.0 represents bottom boundary layer stress in terms of a quadratic bottom drag, which requires the selection of a constant bottom drag coefficient and a velocity scale (section 2.2.4 in Adcroft et al. 2019), just as in the analytical model here. Thus, high-resolution climate models like CM4.0 don't offer an escape from having to place subjective numerical values on model parameters for representing important physical processes.

4. Even if they resolved the basic physics, currently available high-resolution global climate model solutions wouldn't allow for more unambiguous causal attribution. I would still be restricted to a statistical analysis of relations between sea level and streamflow, which would carry the same caveats as the observation-based correlations and regressions presented in the paper. In other words, to rigorously quantify sea-level changes due to streamflow variation would require forcing perturbation experiments where discharge is alternately turned on or off in a given run (Chandanpurkar et al., 2022). Including such a modeling analysis would require coordinating numerical simulations with modeling centers, which is beyond the scope of this study.

While I haven't undertaken the modeling exercise suggested by the reviewer, I acknowledge their points in the revised discussion section by including the following sentence advocating for future studies to perform such experiments:

- *In the future, as global climate models with improved representation of the coastal ocean and shelf seas become more widely available, numerical experiments could be performed to broadly test analytical predictions made here regarding relationships between sea level and streamflow.*

**Technical corrections**:

**R1.CG** l. 48: centred

The sentence is written in the present tense and so the word "center" is preferred to "centred".

**R1.CH** fig3 caption: "thick black" line is not seen

Thanks for catching the typo. It's been corrected in the revision.

**R1.CI** l. 149-150: it should be Sections 4.1, 4.2 and 4.3

Thanks for catching the typos. They've been corrected in the revision.

**R1.CJ** l. 249: should be sections 4.1 and 4.2

Thanks for catching the typos. They've been corrected in the revision.

**References**

- Adcroft, A., et al., 2018, http://hdl.handle.net/1721.1/117188.

- Adcroft, A., et al., 2019, https://doi.org/10.1029/2019MS001726.

- Chandanpurkar, H. A., et al., 2022, https://doi.org/10.1029/2021MS002715.

- Holt, J., et al., 2017, https://doi.org/10.5194/gmd-10-499-2017.

- O'Connor, W. P., 1991, https://doi.org/10.1016/0278-4343(91)90023-Y.

- Piecuch, C. G., et al., 2018, https://doi.org/10.1073/pnas.180542811.

- Piedra-Cueva, I., and M. Fossati, 2007, https://doi.org/10.1016/j.apm.2005.11.033.

---

## Author Comment (AC2)

Response to Reviewer 2
*River effects on sea-level rise in the*
*Río de la Plata during the past century*

by Christopher G. Piecuch

**\*\*\*Reviewer's comments in black\*\*\***
**\*\*\*Author's responses in red\*\*\***

Dear Reviewer 2,

Thank you for your comments on my manuscript. They made the paper stronger, clearer, and more precise. I revised the manuscript based on your reviews, and point-by-point responses follow below. Thank you for your time and consideration.

Best regards,

Christopher G. Piecuch
* * *
**Reviewer 2**—

**General comments**: The manuscript displays the role of the streamflow in the sea level variability especially at long-term trends in the Río de la Plata estuary. To fulfill the objective, annual data from tide gauges and stream gauges are analyzed. The main results indicate that the streamflow is not negligible in the sea level variability and in the long-term trend, except in the south of the river mouth. The river effect increases from the lower estuary to the upper estuary, explaining almost the 60% of the sea level variance. To corroborate that the streamflow is responsible of a percentage of the sea level trend, the author developed a theoretical model finding a coherence between the simulated/predicted data and the observations

The work presented is a hot topic from the climate change point of view. To understand the forcings of the sea level rate in coastal and regional areas is extremely important to prevent and mitigate the consequences. The work also contributes to the analysis of unexplored region compared with other part of the world. Most of the studies in the Río de la Plata estuary were focused on the analysis of the plume dynamics from synoptic to interannual temporal scales using models (e.g: Meccia et al., 2009; Dinapoli et al., 2021; Bodnariuk et al., 2021) and satellite data (e.g., Saraceno et al. 2014). Only a few works showed the sea level rate, however, the causes of the trends were not fully investigated.

Regarding the presentation quality, the manuscript is well-written and well organized. The figures and tables represent the results written.

I thank the reviewer for their positive review. Below they will find responses to all their comments.

**Specific comments**:

**R2.CA** Title: I suggest adding "Estuary" after "Plata"

I will add the word "Estuary" to the revised title.

**R2.CB** 3. Results: Taking advantage of a long sea level record, I suggest studying the acceleration of the sea level rate and the possible relationship with the streamflow, especially in Buenos Aires. The bibliography cited in the manuscript indicates that the sea level is increasing, however, the analysis of a possible acceleration has not been published in the study region.

This comment relates to **R2.CC** and **R2.CD** below. I agree that the length of the time series motivates a more detailed investigation as a function of timescale. For that reason, the revised paper includes a coherence analysis that quantifies the relationship between streamflow and sea level as a function of frequency band (see response to **R2.CD** below). However, estimating sea-level acceleration from tide-gauge data is nontrivial, and results can depend sensitively on time period (Haigh et al., 2014) and acceleration model (Bos et al., 2014; Visser et al., 2015). A meaningful, robust quantification of sea-level acceleration requires an analysis beyond the scope of this study, and is deferred to dedicated future investigations.

**R2.CC** Pag. 4, line 119: see comment on Conclusions

See my response to **R2.CD** immediately below.

**R2.CD** 5.Conclusions Pag. 11, line 290: The author mentioned that the river effects on sea level are apparent at multidecadal and centennial periods. However, I did not find convincing evidence on the paper. There is a discussion based on bibliography about the ENSO signal, the author calculated the correlation between ENSO index and in situ data, and the standard deviation of the streamflow but I was expected a spectral analysis (e.g., wavelet) to asseverate that other signals are also important. For example, it would be interesting to analyze the cross wavelet transform between streamflow and sea level measurements. Regarding the ENSO as an interannual variability, Bodnariuk et al. (2021b) analyzed the effect of SAM (Southern Annular Mode) on the Río de la Plata using a reanalysis model (35-years). The influence of SAM on the sea level was also studied in a wider region including the Mar del Plata tide gauge location (Bodnariuk et al., 2021a; Lago et al., 2021).

To address the reviewer's concern, the revised manuscript includes a coherence analysis quantifying the relation between streamflow and sea level as a function of timescale, which includes the following new text at the end of section 3

- *Streamflow is significantly coherent with sea level at Buenos Aires across most time periods and frequency bands resolved by the data, and with sea level at Montevideo for particular periods and frequencies (e.g., decadal scales generally, interannual scales during the 1990s and 2000s), but mostly incoherent with Mar del Plata sea level (Figure 8).*

and new Figure 8. However, since the paper focuses mainly on sea-level physics, and since previous studies have explored the relationships between streamflow

[Figure]

Figure 8: *Magnitude-squared wavelet coherence between streamflow and sea level at (a.) Buenos Aires, (b.) Montevideo, and (c.) Mar del Plata. Solid black lines identify where values are significant at the 68% confidence level and white dashed lines mark the cone of influence. Statistical significance is determined from 1 000 simulations with phase-scrambled versions of the observational data (Theiler et al., 1992). Values are based on the analytic Morlet wavelet.*

or sea level and large-scale climate, I haven't pursued a more detailed analysis of correlations between streamflow, sea level, and climate modes other than ENSO, as it would be tangential to the paper's primary focus.

**Technical corrections**:

**R2.CE** Replace "Section 4.a" and "4.b" with "4.2" and "4.3"

Thanks for catching the typos. They've been corrected in the revision.

**R2.CF** Figure 3 caption: the colors of the thick lines of Río de la Plata, Río Paraná and Río Uruguay do not match with the legend of the time series.

The thick lines match the legend labels. The thick blue, red, and yellow lines are the streamflow from the Río de la Plata, Río Paraná at Timbúes, and Río Uruguay composite as described in the text and identified in the legend.

**R2.CG** Figure 4 caption: the line styles of the time series do not match with the legend.

Line styles match the legend and no changes to the figure are needed. Note that, when data from only one gauge record is available, the virtual-station record (thin solid) is identical to and sits on top of the record from the gauge with data (thick dashed or thin solid dotted).

**References**

- Bos, M. S., et al., 2014, https://doi.org/10.1093/gji/ggt481.

- Grinsted, A., et al., 2004, https://doi.org/10.5194/npg-11-561-2004.

- Haigh, I. D., et al., 2014, https://doi.org/10.1038/ncomms4635.

- Theiler, J., et al., 1992, https://doi.org/10.1016/0167-2789(92)90102-S.

- Visser, H., et al., 2015, https://doi.org/10.1002/2015JC010716.

---

## Author Response (AR2)

Response to Editorial Comments
*River effects on sea-level rise in the*
*Río de la Plata during the past century*

by Christopher G. Piecuch

**\*\*\*Editor's comments in black\*\*\***
**\*\*\*Author's responses in red\*\*\***

Dear Dr. Williams,

    Thank you for the editorial comments. I addressed all the points you raised. Please see below for more details. Thanks again for your time and consideration.

Best regards,

Christopher G. Piecuch
* * *
**Editorial Comments**—

Thank-you for your revisions. I think you have addressed all of the reviewer's suggestions with one exception (below) and the paper is nearly ready for publication.

I agree that the reviewer's request to compare to OGCMs is beyond the scope of this paper. Tests could perhaps be done with regionally nested models with higher resolution on the estuary, but it would still require a lot more work, and would add complexity to the paper. It would be better done separately.

A few presentational points:

Thanks for the positive assessment. All of your points are addressed below.

**EP1** Fig 1—it would be nicer with the scale in m, and a log colorscale. I think you can do this in Matlab with `set(gca,'colorscale','log')`. For this reason I've selected "minor revisions" rather than "technical", so you have an opportunity to tweak it. Up to you though, I don't insist on it.

Good suggestion. Units are now m and scale is nonlinear.

**EP2** Lines 45-60 the text is a bit clunky, perhaps because it slips into the passive voice. Elsewhere it is clear, thanks.

I changed the wording in a few places in this paragraph to the active voice to make the text less clunky.

**EP3** line 124: "effect" should be "affect" in this instance.

Made the change. Thanks for catching the typo.

**EP4** line 325: A really important point, that you might like to include (in a shorter form) in the abstract.

Done.